# Simulating international tax designs on sugar-sweetened beverages in Mexico

**Juan Carlos Salgado Hernández**[1,2], **Shu Wen Ng**[2,3]*

1 Center for Research in Health Systems, National Institute of Public Health, Cuernavaca, Mexico,
2 Carolina Population Center, University of North Carolina at Chapel Hill, Chapel Hill, North Carolina, United States of America, 3 Department of Nutrition, Gillings School of Global Public Health, University of North Carolina at Chapel Hill, Chapel Hill, North Carolina, United States of America

* shuwen@unc.edu

## Abstract

In response to the high prevalence of overweight and obesity, Mexico implemented a volumetric tax of one Mexican peso (MP) per liter of sugar-sweetened beverage (SSB) in 2014. In contrast to Mexico's volumetric tax design, the United Kingdom (UK) and South Africa (ZA) implemented SSB taxes based on sugar density. This kind of tax is likely to yield larger health benefits than volumetric taxes by imposing a larger tax burden on high-sugar SSB and/or encouraging reformulation. However, sugar-density taxes might yield lower tax revenues. This study aims to simulate the effect of sugar-density taxes as those in the UK and ZA on SSB purchases (in terms of volume and sugar), SSB prices, and tax revenue in Mexico and compare this effect to its counterpart under the current volumetric SSB tax. Additionally, we simulate the effect of sugar-density taxes under different scenarios of reformulation. We conducted all these simulations based on a structural model of demand and supply using household purchase data for 2012–2015 in urban Mexico. We found that the current volumetric one-MP tax led to an SSB purchase reduction of 19% for both volume and sugar and an SSB price increases by MP $1.24. We simulated similar effects under the UK and ZA sugar-density taxes when these taxes were equivalent to the volumetric one-MP tax, and there was no reformulation. When assuming reformulation, the sugar reduction under the sugar-density taxes was up to twice larger than the volumetric one-MP tax. However, we found that the volumetric one-MP tax yielded the largest tax revenue across all tax designs. From a public health perspective, sugar-density taxes are likely to be more effective in tackling the overweight and obesity prevalence in Mexico; however, tax revenue might be lower under these taxes.

## Introduction

By 2012, more than a third of children and teenagers and 71% of adults in Mexico were overweight or had obesity following a sustained increase since 2000 [1, 2]. Concurrently, non-communicable diseases associated with overweight and obesity, such as diabetes mellitus, have risen and thus contributing to large but preventable losses in health and wellbeing [3]. Additionally, the Mexican Ministry of Health estimates that overweight, obesity, and its related non-communicable diseases (NCD) account for 34% of the national health expenditure or 1% of gross

**Data Availability Statement:** We are unable to share the de-identified proprietary third-party dataset due to contractual data use agreements with the data vendor, The Nielsen Company, for its Mexico Consumer Panel Service (CPS) data

(https://www.nielsen.com/us/en/client-learning/consumer-panel-services/). We did not have any special access privileges to the data (we paid for a data license) and got it through the data vendor, the Nielsen Company. For inquiries, please see: Consumer Panel Services – Nielsen (https://www.nielsen.com/us/en/client-learning/consumer-panel-services/).

**Funding:** Our study was funded primarily by Bloomberg Philanthropies (grants to the Carolina Population Center and the Instituto Nacional de Salud Publica), with support from the National Institutes of Health (NIH) (grant number R01DK108148), the Robert Wood Johnson Foundation (Grant No. 71698), and the Carolina Population Center's NIH Center Grant (grant number P2C HD050924). The funders had no role in study design, data collection and analysis, decision to publish, or preparation of the manuscript.

**Competing interests:** The authors have declared that no competing interests exist.

domestic product [4]. Thus, the economic and social costs of overweight and obesity are borne by individuals, their families, and society. While there are several causes of obesity and diabetes, sugar-sweetened beverages (SSB) represent a specific risk factor [5–7]. In 2012, SSB contributed more than two thirds of the overall added-sugar intake to the Mexican diet (12.5% of overall energy intake), exceeding the World Health Organization's sugar limit guidance [8].

To help temper the high SSB consumption and curb the rise in obesity and diabetes in Mexico, the government implemented since January 2014 a specific volumetric excise tax of one Mexican peso (MP) per liter of SSBs with any positive added sugar content (herein "volumetric one-MP tax"). In urban settings, SSB purchase reductions ranged between 8% and 17% in 2014–2015 [9, 10] coupled with price SSB increases in 2014 by at the least the tax amount, i.e., one MP [11]. Building upon the evidence of SSB purchase drops after the tax implementation, Basto-Abreu et al. [12] predicted meaningful population health improvements attributable to the SSB tax that will yield health care savings three times larger than the tax implementation cost. Since Mexico became the first country in the Americas implementing an SSB tax as a national health policy, other countries have enacted such taxes [13].

More recent national SSB taxes have used sugar density (i.e., grams of sugar per 100ml of drink) as the basis for determining the amount of the tax (herein "sugar-density taxes") since the overconsumption of sugar is the underlying issue. Previous studies predict that taxes that target sugar, calories, or other nutrients of concern rather than overall categories of beverage or food are more effective in reducing the consumption of sugar or other harmful nutrients by encouraging substitution towards less sugary or less fattening products [14, 15]. Thus, health benefits are likely to be larger under sugar-density taxes compared to volumetric taxes. In this regard, Grummon et al. [16] assessed comparable volumetric and sugar-density SSB taxes in the United States. They predicted that the latter tax design would lead to a larger sugar intake reduction and more pronounced reductions in obesity prevalence and diabetes incidence [16]. In addition to the potential substitution effect among consumers from high- to low-sugar options due to higher taxes based on sugar-density, this kind of tax might also encourage reformulation by producers aiming to reduce the tax burden to which their products are subject. Conversely, the current volumetric one-MP SSB tax in Mexico is unlikely to lead to reformulation since the tax burden is independent of the sugar content across any SSB with added sugar. In the context of a sugar-density tax for desserts in France, Allais et al. [17] predicted that the tax effect on caloric-sweetener intake from tax-targeted products might be underestimated by up to ≈40% when the potential reformulation induced by the tax implementation is not accounted for.

Among the countries with current sugar-density tax designs for SSB, there are variations in the sugar-density thresholds or tiers used. For example, in April 2018, South Africa (ZA) implemented the Health Promotion Levy (HPL) composed of a new SSB tax, with a linear tax rate per reconstituted liter of SSB at an additional 0.021ZAR per gram of sugar-density beyond 4g (threshold-linear sugar-density) [18]. The United Kingdom (UK) also implemented in April 2018 its Soft Drink Industry Levy (SDIL) using sugar tiers at 5g and 8g per 100ml with a specific rate of £0.18 per liter for drinks with 5-8g sugar-density and £0.24 per liter for drinks with ≥8g sugar-density (multi-tiered sugar-density) [19]. Available evidence suggests that these sugar-density taxes in ZA and the UK have led to product reformulation. In ZA, there is a strong indication that HPL resulted in reformulation given that the sugar purchase reduction (51% lower compared to counterfactual) was larger than the volume purchased reduction (29% lower compared to counterfactual) after the tax implementation [20]. In the UK, the share of drink products subject to SDIL was 33.8 percentage points lower (compared to a counterfactual) within a year of the SDIL implementation [19].

While Mexico's volumetric one-MP tax has been shown to be effective in reducing SSB purchases [9, 10] and intake [21], it remains an open question whether alternative SSB tax designs

as those in other countries could be more effective in reducing SSB volume or sugar consumption in Mexico. Simply taking findings from evaluations conducted in the UK and ZA and assuming they would produce equivalent changes in Mexico if such designs had been used in place of Mexico's volumetric one-MP tax is naïve. This is because the beverage market structure, pre-tax consumption levels and preferences, demographics, and the socio-economic context can be very different across countries.

This paper seeks to more appropriately assess how SSB taxes equivalent to tax designs as in the UK and ZA would change demand-side behaviors (by households via shifts in purchases given their objectives of maximizing utility/happiness) and supply-side responses (by manufacturers via their prices given their objectives of maximizing profit) within the same context (Mexico). We will validate these models by comparing estimates from the structural models under the Mexico's existing one-MP per liter tax policy to observed price changes on the supply-side across beverage types and brands and on the demand side (market shares across beverage types and brands). We will then simulate how alternative tax policies might change SSB purchased in terms of volume and sugar, and thus resultant tax revenues. Then we can compare these various tax policies within the Mexican context. In addition to the price response to the tax by producers, we also aim to account for the potential reformulation under sugar-density taxes assuming different exogenous reformulation scenarios following the approach by Allais et al. [17]. This work provides an innovative approach to allow policymakers to compare SSB tax design alternatives while factoring in how beverage companies (supply-side) and consumers (demand-side) respond. It also builds important groundwork for future efforts to estimate the health and cost-benefit implications across different tax designs.

## Material and methods

### Data

The main source of information for our analysis is the Nielsen Mexico Consumer Panel Service (Nielsen CPS) [22] from 2012 to 2015 so that we have access to data for both pre-tax periods (i.e., 2012–2013) and post-tax periods (i.e., 2014–2015). Nielsen CPS [22] collects information on packaged food and beverage purchases and prices from a household panel and is representative of Mexican urban settings with a population larger than 50,000 inhabitants. Information from households in our analytical data were fully anonymized/de-identified before we received this data. We focused on the non-dairy and non-alcoholic beverage market composed of plain water and diet SSB, which by design do not contain sugar and represent the untaxed beverages, and SSBs with any level of added sugar and thus subject to the tax. We excluded dairy products because of missing information from January 2012 to September 2012. Moreover, we did not focus on tap water because this option is not dominant in urban Mexico due to concerns/mistrust around potability, with only about 10% of households directly reporting tap water as their main drinking water source [23] (this proportion can reach 20% at the national level [24]). We defined a market as the monthly overall beverage purchases in urban Mexico by aggregating households' purchases while accounting for their survey weights. The aggregation of household data at the market level is a common approach in other studies on logit demand models analyzing several years of data (e.g., see Bonnet and Requillart [25], Liu, Lopez, and Zhu [26], and Zhu, Lopez, and Liu [27]). We are interested in analyzing the full data from 2012 to 2015 because we exploit the SSB price change after the tax implementation in January 2014 (see S1 Fig) as the main source of variation to identify the coefficients on prices. We focused on the top 30 products according to their overall purchases in the pre-tax period 2012–2013. These products made up around 96% of the non-dairy and non-alcoholic beverage market in the pre-tax period. The top 30 products include a set of

untaxed beverages mainly composed of 20-liter jars of plain drinking water by local producers, which usually are home-delivered. These beverages represent the outside option in our demand model, as explained below. Excluding the outside option, the remaining brands are produced by seven different firms. We complemented purchase and price information from Nielsen CPS with information on time-invariant sugar content at the SSB brand level from the UNC Mexican Nutrition Fact Panel [28]. This panel collects the most recent available information from products' pictures in different datasets and product-specialized websites (e.g., Redcap) spanning different time periods, as explained elsewhere [10]. We used sugar content at the product level as an SSB's attribute of interest for our demand model. For products in our analytical data with variations in flavors such as flavored waters and juices, we used their average sugar content.

Additional data for our analysis come from the National Institute of Statistics and Geography (termed as INEGI in Spanish). From the National Survey of Occupation and Employment by INEGI [29], we used its quarterly rounds in 2012–2015 to retrieve information on per capita labor income and presence of kids (<13 y) at the household level in urban Mexico, which will work as price utility shifters in our demand model. For households with null monthly labor income, we set an income equal to one to be able to calculate their log income. We exclusively use this household information to approximate one step of the demand model estimation, as explained in the next section. Moreover, we also retrieved from INEGI [30, 31] information on the monthly producer price index for sugar and monthly prices at the product level, which will work as instrumental variables as explained below. Finally, we used the monthly urban consumer price index by INEGI [31] to calculate both real prices and income in January 2014 values.

## Demand model

Following previous structural models on SSB taxes [26, 32, 33], we implement a random-coefficients logit demand model [34, 35]. The indirect utility function for household $i \in (1,...,H)$ when consuming beverage $j$ from the choice set $j \in (0,...,J)$ at market $t \in (1,...,T)$ is:

$$u_{ijt} = \beta Sugar_j + (\sigma_v v_i + \Pi_D D_i)p_{jt} + \xi_j + \Delta\xi_{jt} + \varepsilon_{ijt} \qquad (1)$$

where $Sugar_j$ is beverage $j$'s sugar content in terms of grams per liter, and $p_{jt}$ is beverage $j$'s price per liter at market $t$. $\xi_j$ is beverage $j$'s fixed effects that capture time-invariant unobserved characteristics. For the sake of simplicity in notation, we specify $\xi_j$ at the product level, however, these fixed effects enter the model as brand fixed effects. Eight out of the 29 products in the inside option correspond to brands with light and regular versions that allowed us to identify the coefficient on sugar. $\Delta\xi_{jt}$ corresponds to the beverage $j$'s unobserved characteristics that are time-variant. The utility from prices will vary by both households' unobserved characteristics $v_i$ and observed characteristics $D_i$. We assume that $v_i$ follows a normal distribution with mean zero and standard deviation $\sigma_v$. $D_i$ is a vector of household $i$'s observed characteristics and $\Pi_D$ contains their associated vector of parameters. In our analysis, $D_i$ is composed of two variables. One variable is the log of the household's per capita income and the second is a dummy variable for the presence of kids at the household level. $\varepsilon_{ijt}$ is a type-I extreme value error that is independent and identically distributed across consumers, beverages, and markets.

When the utility of the outside option (i.e., $j = 0$) is normalized to zero, the market share for beverage $j$ at market $t$ is:

$$S_{jt}(p_t, X; \theta) = \int_{H_{jt}} \frac{\exp(\beta Sugar_j + (\sigma_v v_i + \Pi_D D_i)p_{jt} + \xi_j + \Delta\xi_{jt})}{1 + \sum_{j=1}^{J} \exp(\beta Sugar_j + (\sigma_v v_i + \Pi_D D_i)p_{jt} + \xi_j + \Delta\xi_{jt})} dP_D(D)dP_v(v) \qquad (2)$$

where p is a price vector, X is a matrix of beverages' non-price characteristics, $\theta$ is the vector of

structural demand parameters from Eq (1), and $H_{jt}$ represents the set of consumers that get the highest utility from consuming the beverage $j$. Price information in vector p is in terms of the difference of the price of the outside option in market $t$. Households' observed and unobserved characteristics are drawn simultaneously from the respective distributions $P_D(D)$ and $P_\nu(\nu)$. Due to Eq (2) lacks a closed-form solution, the integration is approximated as in Eq (3):

$$S_{jt} = \frac{1}{N}\sum_{i=1}^{N} \frac{\exp(\beta Sugar_j + (\sigma_\nu \nu_i + \Pi_D D_i)p_{jt} + \xi_j + \Delta\xi_{jt})}{1 + \sum_{j=1}^{J}\exp(\beta Sugar_j + (\sigma_\nu \nu_i + \Pi_D D_i)p_{jt} + \xi_j + \Delta\xi_{jt})} \qquad (3)$$

We set an N equal to 1,000 that is the number of draws of $\nu_i$ from $P_\nu(\nu)$ and $D_i$ from $P_D(D)$. In practice for $D_i$, we randomly drew three sets of 1,000 urban households from each quarterly round of the National Survey of Occupation and Employment. We merged each household set to specific monthly markets in Nielsen CPS. We estimate the set of structural parameters $\theta$ of the demand model using the generalized method of moments developed by Berry, Levinson, and Pakes [34] through its Stata routine [36].

### Price endogeneity

A price endogeneity problem is likely to arise due to beverage $j$'s unobserved components in $\Delta\xi_{jt}$, such as advertising, are likely to be related to $p_{jt}$. We use instrumental variables to address this endogeneity problem. Following studies on tobacco assessing tobacco use and prices, where authors instrument prices with exogenous changes in tobacco taxes [37, 38], our first instrument is the exogenous SSB tax implementation since January 2014. We operationalize this tax implementation through a dummy variable to identify all post-tax periods. A threat to the validity of this instrument is that the SSB tax might lead to an awareness change in the negative health effects linked to SSB, and thus a non-price effect might drive the SSB purchase reductions. Teng et al. [39] suggested this could be the case for the SSB tax in Chile, where SSB reductions were larger among well-off households. We cannot properly ascertain if there is indeed a signaling effect and to what degree, but the potential signaling effect is likely to play a smaller role compared to the price effects in Mexico since previous quasi-experimental assessments show that, after the SSB tax implementation, poor households decreased their SSB purchases (both in absolute and relative terms) to a larger extent compared to well-off households [9]. These findings are consistent with previous evidence of larger price-sensitive demands for SSB among more socio-economic-deprived households in Mexico [40].

Following previous studies on SSB and random-coefficients logit demand models [25, 32], our second instrumental variable is a set of interactions between producers and input costs to account for the potential differential production costs across producers, which are supposed to impact beverage prices. Specifically, we interacted producers' identifiers with the monthly producer price index for sugar. Changes in the price index for sugar will work as a proxy of shocks on the production costs of SSB linked to sugar content, leading to respective SSB price adjustments. Thus, we consider that these changes in the price index for sugar must be independent of the beverage $j$'s unobserved components in $\Delta\xi_{jt}$.

For sensitivity analyses of the demand model, we use 2,000 and 5,000 draws of households' characteristics for Eq (3) to test for the precision of the estimated coefficients. Moreover, we also re-run the demand model including seasonal dummies following the equivalent approach in the studies by Zheng et al. [33] and Liu et al. [41]. Finally, we extend the set of instruments by including average monthly prices of beverage $j$ in cities that are not part of Nielsen CPS and correspond to urban locations from which INEGI [31] collects price information to calculate the consumer price index. When a product's price in Nielsen CPS does not have a counterpart in INEGI, we matched it to the average monthly price of products with the same tax status and

by the same producer, which was the case of around 18% of observation in our dataset. This price-related instrument exploits the potential correlation among prices of the same product across cities due to common production costs and the assumption that product valuations are independent across cities [42]. This price-related instrument is relevant to identify the price response of both taxed and untaxed products.

## Supply model

We assume that the multi-product firm $f$ from the set of firms $f \in (1,\ldots,F)$ competes under a Bertrand-Nash supply model. Eqs (4) and (5) respectively represent the firm profit function and the first-order conditions under which firm $f$ maximizes profit.

$$
\begin{aligned}
\pi_f &= \sum_{j \in J_f} (p_{jt} - \tilde{m}c_{jt}) * Q_t * S_{jt}(\mathrm{p}_t, X, \theta) \\
&= \sum_{j \in J_f} (p_{jt} - Tx_{jt} - mc_{jt}) * Q_t * S_{jt}(\mathrm{p}_t, X, \theta)
\end{aligned}
\tag{4}
$$

$$
\begin{aligned}
S_{jt}(\mathrm{p}_t, X; \theta) + \sum_{j \in J_f} T_f(l,j)(p_{jt} - \tilde{m}c_{jt}) \frac{\partial S_{lt}(\mathrm{p}_t, X; \theta)}{\partial p_{jt}} &= 0 \\
S_{jt}(\mathrm{p}_t, X; \theta) + \sum_{j \in J_f} T_f(l,j)(p_{jt} - Tx_{jt} - mc_{jt}) \frac{\partial S_{lt}(\mathrm{p}_t, X; \theta)}{\partial p_{jt}} &= 0
\end{aligned}
\tag{5}
$$

For beverage $j$ in terms of liters, we separate $\tilde{m}c_{jt}$ into the marginal cost $mc_{jt}$ and the tax $Tx_{jt}$ to which the beverage is subject. For post-tax periods and taxed beverages, $Tx_{jt}$ is equal to one in line with the volumetric one-MP SSB tax, and zero in pre-tax periods across all beverages. $S_{jt}(.)$ is the estimated market share for beverage $j$, and $Q_t$ is the overall beverage purchases at the market level. $J_f$ represents the subset of products produced by firm $f$. $T_f(r,j)$ is a matrix whose elements are one when products $l$ and $j$ are produced by the same firm and zero otherwise. $\partial S_{lt}(.)/\partial p_{jt}$ corresponds to the firm reaction matrix where each element captures the change in beverage $l$'s market share when beverage $j$'s price changes take place. We solve for $mc_{jt}$ in Eq (5) to recover marginal cost information that is a component of the simulation process under different tax designs, as explained below.

## Tax designs and simulations

In addition to the assessment of the current tax of one MP per SSB liter, we analyze the effect of equivalent SSB taxes as in the UK and ZA in the Mexican setting. In the UK, the multi-tiered sugar-density tax design exempts SSBs with <5 sugar grams per 100 ml from the levy. In contrast, SSB are subject to a low levy (£0.18 per liter) when their sugar content ranges between 5 and 8 sugar grams per 100 ml, and a high levy (£0.24 per liter) when SSB overpass eight sugar grams per 100 ml [19]. In ZA, the threshold-linear sugar-density tax design levies an SSB tax of 0.021 ZAR to each sugar gram beyond the first four grams of sugar density [18].

We define two sets of taxes to analyze the UK and ZA tax designs in the Mexican setting. The first set corresponds to tax amounts equivalent to the proportion of the UK and ZA taxes compared to their pre-tax prices. These tax amounts for the Mexican setting are MP$0.45 and MP$0.81 per liter for the low- and high-levy SSB as in the UK tax design, respectively, and MP$0.014 per gram of sugar density beyond 4 grams as in the ZA tax design. We refer to this set of taxes as "equivalent to international taxes". The second tax set is composed of tax amounts that will increase average prices by one MP in case of a full tax pass-through. Under this

approach, the tax is MP$0.755 and MP$1.01 per liter for the low- and high-levy SSB as in the UK tax design, respectively, and MP$0.0156 per gram of sugar density beyond 4g under the ZA tax design. Thus, the second tax set allows us to make a direct comparison between the current volumetric one-MP tax and the alternative UK and SA tax designs since the tax burden is the same across all these SSB tax designs. We refer to this second set of taxes as "equivalent to the one-MP tax". In the S1 File, we explain the procedure to calculate these sets of SSB taxes in the UK and ZA for the Mexican setting and illustrate the Mexico (red), UK (blue), and ZA (green) designs under the two tax sets described above in Fig 1.

We use Eqs (6) and (7) to simulate the equilibrium prices at the market level under each tax design of interest. Eq (6) presents the first-order conditions under which firms re-maximize profit in response to the tax design of interest, and Eq (7) is the objective function for the optimization process

$$S_{jt}(p_t,\ X;\theta) + \sum_{j\in J_f} T_f(l,j)(p_{jt} - \tilde{mc}_{jt,Ex,c})\frac{\partial S_{lt}(p_t,\ X;\theta)}{\partial p_{jt}} = 0$$

$$S_{jt}(p_t,X;\theta) + \sum_{j\in J_f} T_f(l,j)(p_{jt} - Tx_{jt}^{Ex,c} - mc_{jt})\frac{\partial S_{lt}(p_t,X;\theta)}{\partial p_{jt}} = 0$$

(6)

$$max = [(p_t^{iter}) - (p_t^{iter-1})]^2 < 10^{-6}$$

(7)

In Eq (6), we set $\tilde{mc}_{jt,Ex,c}$ as the sum of $mc_{jt}$ and $Tx_{jt}^{Ex,c}$ where $Tx_{jt}^{Ex,c}$ represents the excise tax for taxed beverage $j$ per liter according to the tax design in country $c$. $Tx_{jt}^{Ex,c}$ for taxed beverages will equal one for the case of the volumetric one-MP tax. For tax designs as in the UK or ZA, $Tx_{jt}^{Ex,c}$ is a function of the beverage $j$'s sugar content. In Eq (7), $p_t^{iter}$ and $p_t^{iter-1}$ are the vector of simulated consumer prices from the iteration iter and the preceding iteration iter−1,

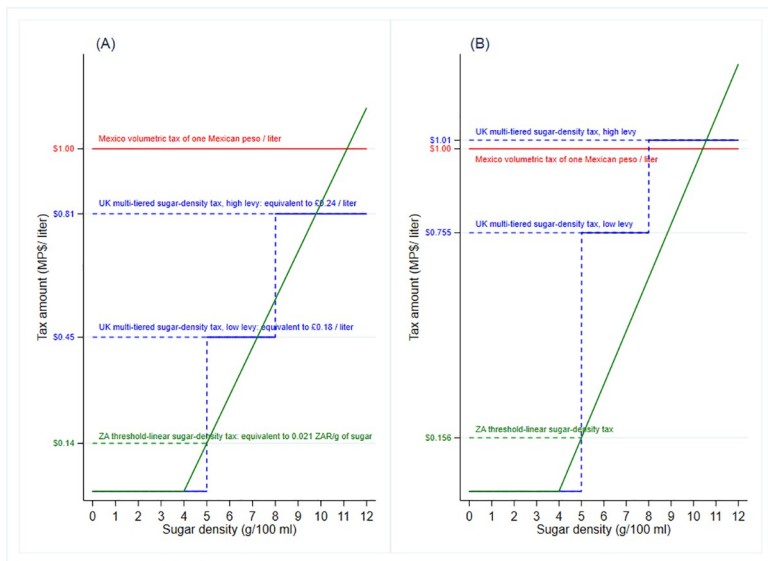

**Fig 1. Two sets of sugar-density taxes compared to the one-Mexican peso tax.** (A)Equivalent to international taxes (B) Equivalent to one-Mexican peso. UK: United Kingdom. ZA: South Africa. *Source*: Authors' own analyses and calculations based on data from Nielsen through its Mexico Consumer Panel Service (CPS) for the food and beverage categories for January 2012 –December 2015. The Nielsen Company, 2016. Nielsen is not responsible for and had no role in preparing the results reported herein.

respectively. We conduct the simulation process by adapting the optimization procedure in Mata by Lorincz [43] based on a quasi-Newton algorithm.

For post-tax periods, we initiate the simulation process by setting all values in $Tx_{jt}^{Ex,c}$ equal to zero in Eq (6) and thus simulate a vector of consumer prices $\tilde{p}_t^{NoTx}$, which represents the equilibrium consumer prices in the absence of the volumetric one-MP tax in Mexico in 2014 and 2015. We use $\tilde{p}_t^{NoTx}$ along with the information in $Tx_{jt}^{Ex,c}$ under each tax design of interest to simulate the consumer prices when the corresponding tax comes into effect. Thus, the tax effect on prices corresponds to the difference between the simulated consumer prices for each tax design and $\tilde{p}_t^{NoTx}$. S1 File shows the procedure that we follow to estimate the tax effect on purchases (volume and sugar) and tax revenue. We report the tax effect as the average difference in the outcome of interest across tax designs but are unable to provide uncertainty ranges of the estimated difference (e.g., standard errors). This approach is standard in simulations based on structural models of demand and supply as the one we used in this study (for example, see Zheng, Huang, and Ross [33], Liu, Lopez, and Zhu [26], and Bonnet and Requillart [25]). Obtaining uncertainty ranges in our estimates through methods such as bootstrapping can result in unrepresentative shares by firms given the sampling procedures and, therefore, it can result in non-convergence of the models as well as implausible values.

## Reformulation

In their study on a sugar-density tax for the dessert market in France, Allais et al. [17] accounted for the response to this tax by consumers (via production substitution) and producers (via price re-maximization). Moreover, they introduced potential product reformulation and assumed this is exogenous. Allais et al. [17] calculate the new products' composition and the resulting changes in marginal costs after reformulation through optimization models and a set of constraints related to products' recipes and ingredients along with product input costs. The authors stated that producers have an incentive to reformulate when the reformulation cost is lower than the tax burden in the absence of reformulation.

In a spirit similar to the approach by Allais et al. [17], we assume a set of exogenous reformulation scenarios; however, rather than modeling products' new composition, we set two extreme reformulation scenarios. For the first reformulation scenario, we assume a partial substitution of sugar for artificial non-caloric sweeteners keeping SSB sweetness levels unchanged. Thus, consumers are supposed to experience no change in SSB flavor, which implies no utility change other than the one induced by the tax effect on prices. Moreover, we assume that the marginal cost increase linked to the reformulation is lower than the tax burden in the absence of the reformulation. For the second reformulation scenario, we assume that producers reduce sugar in SSBs with no substitution for artificial non-caloric sweeteners. As a result of this sugar/sweetness reduction, SSBs will be subject to a lower tax burden; however, consumers will experience a utility loss attributable to the sweetness reduction. From the producers' perspective, the sugar reduction is equivalent to an input reduction, and thus we assume that marginal costs will drop consequently. By setting these two extreme reformulation scenarios, we intend to provide lower and upper bounds of the potential reformulation effect in Mexico. Herein, we refer to the first and second reformulation scenarios as "sweetness unchanged" and "sweetness reduction", respectively.

For the UK tax design under reformulation, we assume that producers reduce sugar density by two grams per 100 ml across those SSB whose sugar density is up to two grams above the next lower tax threshold. Therefore, SSB with an original sugar density of 5–7 or 8–10 sugar grams per 100 ml will be part of the tax categories of no levy or low levy after reformulation, respectively. For the sweetness unchanged scenario, we assume that reformulation induces a

marginal cost increase equivalent to 50% of the tax burden difference from moving down across tax thresholds (e.g., from the high levy to the low levy). In contrast, we assume a marginal cost decrease equivalent to 105% of the tax burden difference from moving down across tax thresholds for the scenario of sweetness reduction. For the ZA tax design under reformulation, we assume a uniform sugar reduction by 30% across SSB with a sugar density above four grams per 100 ml. As for the UK tax design where marginal cost adjustments are proportional to tax burden differences with and without reformulation, we also assume equivalent marginal cost increases by 50% for the scenario of sweetness unchanged and decreases by 105% for the scenario of sweetness reduction. All reformulation scenarios are restricted to the UK and ZA tax designs equivalent to the volumetric one-MP tax. Moreover, within either of the UK or ZA tax designs, we first set each of the reformulation scenarios only across SSB produced by the two major producers. We subsequently set reformulation across all producers. We implement this differentiated reformulation across producers because major producers are more likely to have more resources that allow them to be the first in adapting their production process in response to the sugar-density tax.

For the reformulation scenarios of sweetness unchanged, we adjust our structural model by re-writing $\tilde{mc}_{jt,Ex,c}$ in Eq (6) as the sum of $Tx_{jt}^{Ex,c} + mc_{jt} + \Delta mc_{jt,Ex,c}$. Here, $\Delta mc_{jt,Ex,c}$ stands for the marginal cost increase induced by the reformulation under either of the sugar-density tax designs in the country of interest $c$, i.e., UK or ZA. This marginal cost increase entails that artificial non-caloric sweeteners are more costly than sugar. For supplementary analyses, we assume no differential costs between artificial non-caloric sweeteners and sugar so that $\Delta mc_{jt,Ex,c}$ will equal zero. For the reformulation scenarios of sweetness reduction, we incorporate $\tilde{mc}_{jt,Ex,c}$ as described above and update the sugar information after reformulation in the matrix X of beverages' non-price characteristics in Eq (6). In S1 File, we extend our explanation on the procedures to estimate the tax effect on purchases and tax revenue when assuming or not assuming reformulation.

## Results

### Analytical data descriptives

Table 1 shows the summary statistics of our analytical data. The market share for taxed beverages in Mexico decreased from 19.61% in the pre-tax period (2012–13) to 17.56% in the post-period (2014–15). Likewise, there was a reduction in the market share of untaxed beverages. This simultaneous drop for both taxed and untaxed beverages might result from a purchase power reduction in branded beverages available within the store retail system. In contrast, the outside option, which is mainly composed of untaxed home-delivered water by local producers, increased their market share from 60.11% in the pre-tax period to 64.25% in the post-tax. In line with the overall market share changes, we saw equivalent changes in the per-capita per-day purchases. Table 1 also shows a price increase by ≈ MP$0.90 for taxed beverages and no major change for untaxed beverages in the post-tax period. Additionally, we show the sugar density distribution across beverages in our analytical data in Fig 2. This shows that in Mexico, beverages with no sugar (i.e., untaxed beverages) represent around 30% of the beverages, while sugar density across all SSB is above 4 grams per 100 ml, and most SSBs contain more than 8 grams per 100 ml. Thus, all SSBs will be subject to a tax under the ZA tax design, and most SSBs will be subject to the high-levy tax (i.e., >8 sugar grams per 100 ml) under the UK tax design.

### Demand model estimates and model fit

The demand model estimates in Table 2 show a utility increase linked to beverages' sugar content while an inverse relationship holds between utility and prices. However, this relationship

**Table 1. Summary statistics.**

| | | 2012–2013 (pre-tax) | 2014–2015 (post-tax) |
|---|---|---|---|
| **Market shares** | **Taxed beverages** | 19.61 | 17.56 |
| | **Untaxed beverages** | 20.27 | 18.19 |
| | **Outside option** | 60.11 | 64.25 |
| **PC-PD purchases** | **Taxed beverages** | 182.77 | 165.42 |
| | | (10.50) | (9.81) |
| | **Untaxed beverages** | 189.05 | 171.68 |
| | | (12.66) | (13.93) |
| | **Outside option** | 561.25 | 608.49 |
| | | (42.53) | (64.22) |
| **Prices per liter** | **Taxed beverages** | 8.52 | 9.40 |
| | | (1.52) | (1.44) |
| | **Untaxed beverages** | 2.04 | 2.05 |
| | | (1.92) | (2.04) |
| | **Observations** | 696 | 696 |

*Note*: Prices are calculated as quantity-weighted average prices. Standard deviation in parentheses. PC-PD: Per capita Per day. *Source*: Authors' own analyses and calculations based on data from Nielsen through its Mexico Consumer Panel Service (CPS) for the food and beverage categories for January 2012 –December 2015. The Nielsen Company, 2016. Nielsen is not responsible for and had no role in preparing the results reported herein.

varies across households by composition and income. The small and statistically insignificant standard deviation of the random price coefficient might arise from the lack of price variation. Fox et al. [44] highlight that the identification of the random coefficients relies on the variation in the variables of interest. In our demand model specification, the lack of price variation might result of the use of instruments (i.e., the tax implementation and sugar costs), which

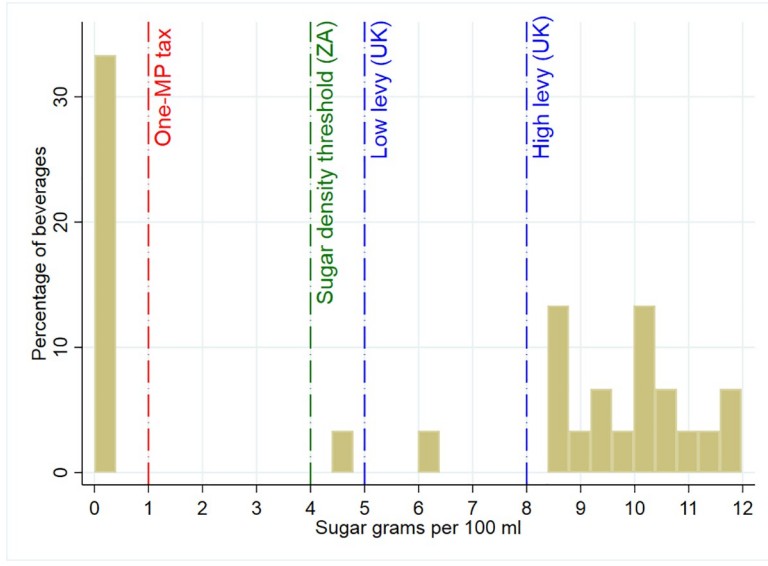

**Fig 2. Sugar distribution across beverages in Mexico.** MP: Mexican peso. UK: United Kingdom. ZA: South Africa. *Source*: Authors' own analyses and calculations based on data from Nielsen through its Mexico Consumer Panel Service (CPS) for the food and beverage categories for January 2012 –December 2015. The Nielsen Company, 2016. Nielsen is not responsible for and had no role in preparing the results reported herein.

impact all SSB and entail a correlation across taxed beverages, and the data aggregation at the urban Mexico level, which overlooks market variations across Mexican cities. At the bottom of Table 2, we show that the instrumental variables to address price endogeneity were strong with an F-stat of around 1,203.8. In S1 Table, we provide the estimates of the first stage where we regressed prices on the set of instrumental variables. Based on the demand model estimates and the first-order conditions for the supply model, we recovered the marginal costs. In S2 Fig, we present the distribution of marginal costs across products and the average marginal costs by the full categories of taxed and untaxed beverages from 2012 to 2015. We found these marginal costs displayed, in general, a stable pattern for the full period of interest.

In S2 Table, we compare the observed and simulated prices and market shares under the volumetric one-MP tax in 2014–15 to assess the model simulation performance. We found no major difference among observed and simulation outcomes for both taxed and untaxed beverages showing that there is a good model fit. This model fit provides support on the reliability of the ex-ante assessment of the alternative SSB taxes in this study. However, it is worth noting that we conducted the simulation process with around 4% of the observations with negative marginal costs (average of MP\$ -0.12). Negative marginal costs mean that for each additional unit of a product, there is a decrease in total costs. These observations were for plain water, so it could be possible that because plain water is a socially desirable good, producers may be getting a subsidy or credit for producing this kind of beverage. For example, one of the leading companies in the non-dairy and non-alcoholic beverage market pays USD \$0.10 for 1,000 liters of water to the Mexican federal government [45]; meanwhile, Mexican households pay about USD \$0.70 for the same amount of water [46].

**Table 2. Estimated coefficients of the demand model.**

|  | Coefficient |
|---|---|
| Mean utility |  |
| Sugar | 0.0185*** |
|  | (0.000423) |
| Price | -1.639*** |
|  | (0.356) |
| Price |  |
| Income | 0.156*** |
|  | (0.0418) |
| Kid | -0.683** |
|  | (0.264) |
| Standard Deviation | 0.00569 |
|  | (0.246) |
| **First stage** |  |
| F statistics | 1203.8 |
| Observations | 1392 |

Note: Standard error in parentheses. Model includes brand fixed effects. Kid stands out for the presence of household members aged <13 years. Results are based on 1,000 draws of the households' unobservable characteristics. + $p <$ 0.10, * $p < 0.05$

** $p < 0.01$

*** $p < 0.001$. Source: Authors' own analyses and calculations based on data from Nielsen through its Mexico Consumer Panel Service (CPS) for the food and beverage categories for January 2012 –December 2015. The Nielsen Company, 2016. Nielsen is not responsible for and had no role in preparing the results reported herein.

The estimated structural paraments from the demand model were robust to an increase in the numbers of draws in Eq (3), the inclusion of the seasonality dummy variables, and the inclusion of prices from other cities as part of the instrumental variables. Moreover, we ran a standard logit demand model accounting for the price endogeneity and no random coefficients for comparability reasons. We found that coefficients on prices and sugar from the random-coefficients demand model were, respectively, larger (in absolute terms) and similar compared to their counterparts in the standard logit demand model. This is consistent with findings by Nevo [42], who found larger estimated coefficients on prices when using random coefficients and comparable estimates for products' attributes with no random coefficients in either model (i.e., random-coefficients demand logit or standard demand logit). The difference in estimates across models in our study might arise from the random coefficients capturing larger dis-utility attached to price increases across households with different characteristics, which by design is overlooked in the standard logit model. We present all demand model specifications for the sensitivity analyses in S3 Table.

## Tax effect on prices, purchases, and tax revenue by tax design

Fig 3 shows the tax effect on SSB purchases in terms of volume and sugar under the UK and ZA tax designs, respectively. We predict an SSB purchase reduction by ≈19% for both volume and sugar under the volumetric one-MP tax, which we include as the red dotted line as the baseline tax design in Fig 3. When there is no reformulation for either of the UK and ZA tax designs, the tax effect on both volume and sugar is lower than the volumetric one-MP tax for international equivalent tax designs and similar for the one-MP equivalent tax designs. However, we observe a slightly larger tax effect on sugar compared to volume across the UK and ZA tax designs with no reformulation. When we assume reformulation for either of the one-MP equivalent tax designs, we observe larger tax effects on sugar than its counterpart in terms of volume or the tax effect under the volumetric one-MP tax. Under reformulation, the UK tax design barely overpasses the tax effect on sugar compared to the volumetric one-MP tax.

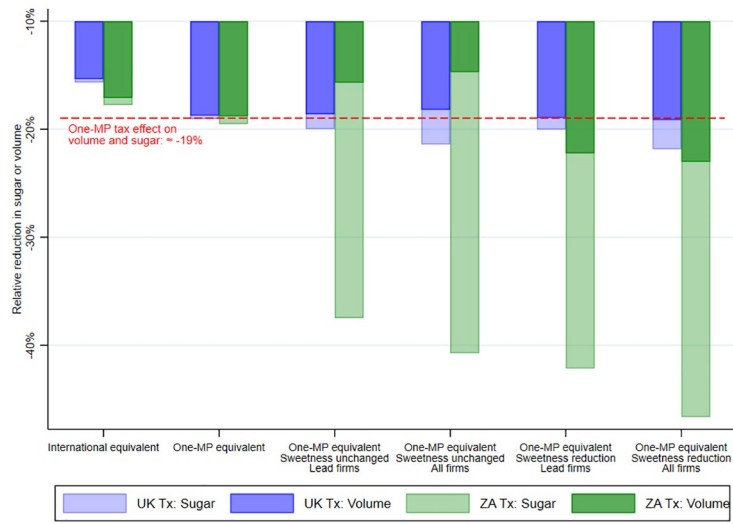

**Fig 3. Tax effect in 2014 and 2015 by tax design.** MP: Mexican peso. UK: United Kingdom. ZA: South Africa. *Source*: Authors' own analyses and calculations based on data from Nielsen through its Mexico Consumer Panel Service (CPS) for the food and beverage categories for January 2012 –December 2015. The Nielsen Company, 2016. Nielsen is not responsible for and had no role in preparing the results reported herein.

Conversely, the sugar reduction for the ZA tax design under reformation ranged between 37% and 47%, which implies a tax effect twice larger compared to the volumetric one-MP tax. In general, we observe larger sugar reductions for the UK and ZA tax designs when all producers reformulate and when we compare the reformulation scenario of sweetness reduction to sweetness unchanged.

In Tables 3 and 4, we present the tax effect under the UK and ZA tax designs on other outcomes of interest, including the implied prices elasticity of demand for volume and sugar, which we calculated as the ratio of the percentage changes as in Fig 3 over the percentage SSB price change. We do not report tax pass-through and price elasticities under reformulation due to simultaneous changes in the tax burden, marginal costs, and sugar content. In both Tables 3 and 4, we present the tax effect under the volumetric one-MP tax as the baseline tax design. Compared to the simulated SSB price in the absence of the tax, SSB prices go up by MP $1.24 for the volumetric one-MP tax. When there is no reformulation, the SSB price increase is similar for the UK and ZA tax designs equivalent to one-MP taxes (as intended); meanwhile, this price increase is below one MP for the international equivalent taxes. All these price increases entail tax over-shifting of the SSB tax regardless of the tax design. Moreover, in line with all these price increases, we observe proportional reductions in SSB per capita purchases in terms of volume and sugar, translating into price elasticities of demand ranging between -1.2 and -1.3. Consistent with Fig 3 under UK and ZA tax designs, the implied price elasticity of sugar tends to be slightly higher (in absolute value) compared to the price elasticity of volume. In S4 Table, we provide further results across SSB depending on their tax rates according to the UK tax tiers. When assuming reformulation across one-MP equivalent taxes, we see price increases below their counterparts in the absence of reformulation, which is consistent with the lower tax burden due to the reformulation. It is worth noting that the lower price increase is remarkable for the ZA tax designs under the reformulation scenario of sweetness

**Table 3. Tax effect in 2014 and 2015: United-Kingdom tax design.**

| Outcomes | No tax in place | Volumetric One-MP Tax | Effect compared to No Tax in place | | | | | |
|---|---|---|---|---|---|---|---|---|
| | | | No reformulation | | Reformulation (One-MP equivalent) | | | |
| | | | International equivalent | One-MP equivalent | Sweetness unchanged | | Sweetness reduction | |
| | | | | | Lead firms | All firms | Lead firms | All firms |
| Prices ($ MP) | 8.17 | 1.24 | 0.99 | 1.22 | 1.21 | 1.18 | 1.18 | 1.10 |
| Prices (%) | | 15.12 | 12.10 | 14.96 | 14.87 | 14.48 | 14.46 | 13.43 |
| Tax Pass-through level (%) | | 123.52 | 124.42 | 123.04 | - | - | - | - |
| PC-PD Volume (mL) | 204.14 | -38.72 | -31.36 | -38.25 | -38.01 | -37.16 | -38.72 | -39.11 |
| Volume (%) | | -18.97 | -15.36 | -18.74 | -18.62 | -18.20 | -18.96 | -19.15 |
| Implied price elasticity of demand (volume) | | -1.25 | -1.27 | -1.25 | - | - | - | - |
| PC-PD Sugar (grams) | 21.30 | -4.04 | -3.34 | -4.06 | -4.26 | -4.56 | -4.27 | -4.65 |
| Sugar (%) | | -18.98 | -15.68 | -19.07 | -19.99 | -21.42 | -20.03 | -21.85 |
| Implied price elasticity of demand (sugar) | | -1.26 | -1.30 | -1.27 | - | - | - | - |
| Tax revenue per capita per year ($ MP) | | 60.38 | 49.77 | 59.77 | 58.87 | 56.74 | 58.78 | 56.19 |

Note: Prices are calculated as quantity-weighted average prices. MP: Mexican Pesos, PC-PD: Per capita Per Day. Source: Authors' own analyses and calculations based on data from Nielsen through its Mexico Consumer Panel Service (CPS) for the food and beverage categories for January 2012 –December 2015. The Nielsen Company, 2016. Nielsen is not responsible for and had no role in preparing the results reported herein.

**Table 4. Tax effect in 2014 and 2015: South-Africa tax design.**

| Outcomes | No tax in place | Volumetric One-MP Tax | Effect compared to No Tax in place | | | | | |
| --- | --- | --- | --- | --- | --- | --- | --- | --- |
| | | | No reformulation | | Reformulation (One-MP equivalent) | | | |
| | | | International equivalent | One-MP equivalent | Sweetness unchanged | | Sweetness reduction | |
| | | | | | Lead firms | All firms | Lead firms | All firms |
| Prices ($ MP) | 8.17 | 1.24 | 1.11 | 1.23 | 1.03 | 0.95 | 0.21 | 0.02 |
| Prices (%) | | 15.12 | 13.63 | 15.08 | 12.67 | 11.60 | 2.59 | 0.21 |
| Tax Pass-through level (%) | | 123.52 | 123.54 | 122.66 | - | - | - | - |
| PC-PD Volume (mL) | 204.14 | -38.72 | -34.93 | -38.39 | -32.06 | -30.02 | -45.45 | -47.08 |
| Volume (%) | | -18.97 | -17.11 | -18.80 | -15.70 | -14.71 | -22.23 | -23.02 |
| Implied price elasticity of demand (volume) | | -1.25 | -1.26 | -1.25 | - | - | - | - |
| PC-PD Sugar (grams) | 21.30 | -4.04 | -3.78 | -4.16 | -7.99 | -8.68 | -8.98 | -9.94 |
| Sugar (%) | | -18.98 | -17.76 | -19.52 | -37.49 | -40.73 | -42.15 | -46.64 |
| Implied price elasticity of demand (sugar) | | -1.26 | -1.30 | -1.29 | - | - | - | - |
| Tax revenue per capita per year ($ MP) | | 60.38 | 54.93 | 59.87 | 36.63 | 32.36 | 33.99 | 29.04 |

Note: Prices are calculated as quantity-weighted average prices. MP: Mexican Pesos, PC-PD: Per capita Per Day. Source: Authors' own analyses and calculations based on data from Nielsen through its Mexico Consumer Panel Service (CPS) for the food and beverage categories for January 2012 –December 2015. The Nielsen Company, 2016. Nielsen is not responsible for and had no role in preparing the results reported herein.

reduction. In light of this low-price increase under this reformulation scenario, it is the sugar reduction what mainly drives the decrease in SSB purchases in terms of volume and sugar for this tax design. In S5 Table, we show the estimates of the reformulation scenario of sweetness unchanged when assuming no differential cost between sugar and artificial sweeteners. These estimates remain similar to those under the assumption of marginal cost increase for the scenario of sweetness unchanged; however, we found slightly lower SSB price increases (as expected) and consequently lower SSB purchase reductions in terms of both volume and sugar. Conversely, we saw an increase in tax revenue, which remains below its counterpart under the one-MP volumetric tax.

Regarding the tax revenue across the tax designs, we found the largest yearly per capita tax revenue under the volumetric one-MP tax, which was equal to MP$ 60.38. In line with the tax effect on purchases in the absence of reformulation, the tax revenue under the international equivalent and one-MP equivalent taxes is lower and similar to the volumetric one-MP tax, respectively. Thus, the one-MP equivalent taxes in the absence of reformulation will lead to a scenario of fiscal neutrality (i.e., same tax revenue) compared to the current volumetric one-MP tax. In contrast, the tax revenue tends to be lower under all reformulation scenarios compared to the volumetric one-MP tax. This lower tax revenue is a consequence of the reformulation that reduces the tax burden at the product level. Thus, we found the largest drops in tax revenue under the ZA tax designs when we assume a reformulation across the full sugar distribution for SSB with a sugar density larger than four sugar grams per 100 ml.

## Tax designs on market shares for volume with no reformulation

In Fig 4, we present the overall market shares in terms of volume under each tax design of interest in 2014–2015 with no reformulation. In the absence of the SSB tax, 60.94% of the

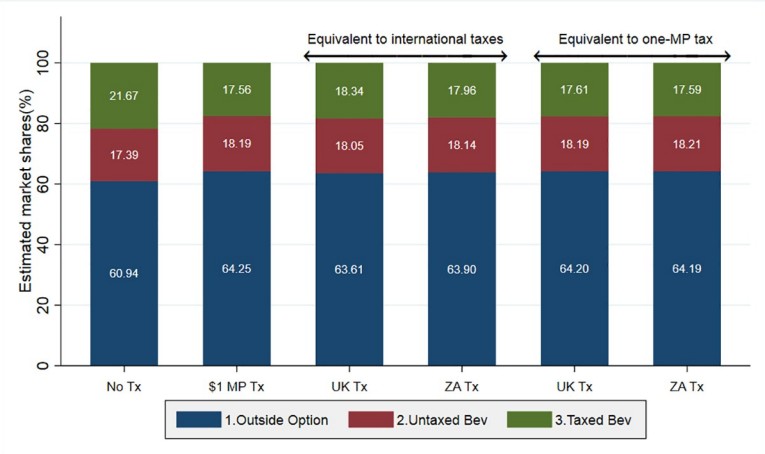

**Fig 4. Estimated market share by beverage type and tax design.** MP: Mexican peso. UK: United Kingdom. ZA: South Africa. *Source*: Authors' own analyses and calculations based on data from Nielsen through its Mexico Consumer Panel Service (CPS) for the food and beverage categories for January 2012 –December 2015. The Nielsen Company, 2016. Nielsen is not responsible for and had no role in preparing the results reported herein.

market corresponds to the outside option, 17.39% to untaxed beverages, and 21.67% to taxed beverages. The market share for the latter kind of beverages goes down to 17.56% under the volumetric one-MP tax. Consequently, the market share for beverages exempted from the SSB tax (i.e., untaxed beverages and the outside option) reaches 82.44%. These market shares prevail under the UK and ZA tax designs when they are equivalent to the one-MP tax. In contrast, when these taxes are equivalent to international taxes, taxed beverages corner around 18% of the market, and thus around 80% of the market corresponds to beverages not subject to the tax. In Fig 4, we do not present the market shares under reformulation since we have already shown above that the reformulation effect is mainly on sugar rather than volume.

## Discussion

In this paper, we applied two sugar-density tax designs from the UK and ZA to data from Mexico to simulate changes in beverage purchases (measured in volume and sugar) and tax revenues, with and without assumptions around reformulation. This approach allows us to compare changes in these outcomes under Mexico's existing volumetric one-MP SSB tax compared to UK's multi-tiered sugar-density design and ZA's threshold-linear sugar-density design. We built all our estimates upon a structural model of demand and supply accounting for the non-dairy and non-alcoholic beverage market structure in Mexico and thus the distribution of sugar-density across beverages. We validated our structural model exploiting the volumetric tax in place since 2014 and showed a proper model fit that should increase the reliability of our simulated tax effects under the different tax designs of interest.

Our simulations show that international equivalent sugar-density taxes will lead to lower reductions in both volume and sugar compared to the respective ≈19% reduction under the volumetric one-MP tax. In contrast, one-MP equivalent sugar-density taxes in the absence of reformulation will perform very similar in terms of volume and sugar reductions compared to the volumetric one-MP tax. Under the reformulation scenarios, one-MP equivalent sugar-density taxes, particularly the ZA tax design, will reduce sugar from SSB to a larger extent compared to the volumetric one-MP tax. However, no tax design outperforms the volumetric one-MP tax in terms of tax revenue.

In this study, we set the exogenous reformulation scenarios for sugar-density taxes as extreme responses by producers to this kind of tax. Thus, we expect that our estimated tax effects under reformulation should provide lower and upper bounds of the potential effects of sugar-density taxes in Mexico. Studies assessing the respective sugar-density taxes in the UK and ZA suggest that this kind of taxes effectively encourages reformulation [19, 20]. However, it is worth noting that in these countries, there was around a two-year window between their respective announcement of the sugar-density tax and the implementation [19, 20]. Thus, this time window should have provided beverage producers ample time and opportunity to cut their products' sugar-density and market their new/reformulated products. That said, it is unclear whether in the context of Mexico, if the government had opted for a sugar-density tax instead of a volumetric tax, reformulations as in the UK or ZA would have occurred given that Mexico's tax law was passed in late-September 2013 and implemented in January 2014 [47, 48].

In addition to the time-span limitation, our study has other limitations. First, our estimated effects across all tax designs of interest are not representative at the country level as the Nielsen CPS [22] data only include information from cities with a population larger than 50,000 inhabitants. Thus, we cannot generalize our findings due to existing differences between rural and urban settings. In terms of market composition, a lower proportion of rural households directly reports bottled water as their main drinking water source [23]. Moreover, there was a lower magnitude of both SSB price increases and SSB purchases drops after the tax implementation in the rural areas compared to urban areas [11, 49, 50]. In light of these differences, we might expect a lower tax effect if we were able to implement our model in rural areas, but without empirical data, this is difficult to know with certainty. Second, in the presence of a signaling effect linked to the SSB tax, we would overestimate the tax effect attributable to price increases. A cross-sectional study based on 2016 data [51] showed that awareness of the SSB tax was more predominant across people from high-income and urban households. Greater awareness of the tax was associated with a higher probability of self-reporting a reduction in SSB consumption. However, the authors stated that this association does not entail causality due to potential unobservable variables linked to the SSB tax awareness [51]. Surprisingly, a larger proportion of people from high-income households aware of the SSB tax believed that the tax was not effective in reducing SSB consumption [51]. In light of these findings and previous evidence showing a negative relationship between SSB purchase reductions and households' socio-economic status [9], consistent with demand price elasticities across households [40], we consider that price increases drove the reduction in SSB purchases. Third, in line with the highlighted limitation by Allais et al. [17] regarding the exogenous reformulation, we acknowledge that our estimated effects under reformulation and the extent of the reformulation might not correspond to market-equilibrium outcomes. Fourth, we only modeled consumers' preferences over sugar rather than artificial/non-caloric sweeteners. Our assumption that consumers would experience no change in SSB flavor and thus utility can likely lead to an overestimation of the tax effects under the reformulation scenario of sweetness unchanged. However, we do not consider this limitation to be restrictive in the context of our study since the purpose of these reformulation scenarios is to provide a range of their potential effects. Fifth, we did not model product entry as a producer response to the sugar-density tax. However, preliminary evidence of the sugar-density tax in ZA suggests that, compared to reformulation, net product entry (i.e., entry minus exit) plays a minor role in the industry response to the sugar-density tax [52]. Sixth, we assumed that artificial sweeteners are more expensive than sugar for the reformulation scenario of sweetness unchanged. Although we consider this assumption to be feasible, we found no reference to support it. However, our general findings and conclusion remain, in general, the same when assuming no differential cost between

artificial sweeteners and sugar. Finally, and as standard in logit demand models, we hold fixed the monthly market size (i.e., $Q_t$), under the different tax simulation scenarios (see equations S.1 and S.2 in S1 File). Thus, we assumed no variation in the overall volume sales of non-dairy and non-alcoholic beverages due to the SSB tax. This assumption implies the full substitution of taxed beverages for either untaxed beverages or the outside option, which seems feasible because these latter options mainly correspond to water.

The strengths of this paper include the use of rich Nielsen CPS data and a validated structural model. Specifically, this structural model accounts for the strategic and simultaneous response to the tax implementation by consumers, who re-maximize utility through substitution, and producers, who define a new set of prices across their full product portfolio to re-maximize profit. We validated our structural model taking advantage of the disaggregated data at the product level in Nielsen CPS that includes pre- and post-tax information. These disaggregated data allowed us to account for the non-dairy and non-alcoholic beverage market structure in Mexico. Thus, we contribute to the literature of the assessment of the volumetric one-MP tax by providing evidence built into both the demand and supply sides. Previous studies on SSB purchase reduction after the implementation of the volumetric one-MP tax have solely focused on the response by consumers and found that this purchase reduction ranged between 8 and 17% [9, 10]. We found SSB purchase reductions for the volumetric one-MP tax slightly above estimates from these previous studies. Specifically, our findings show that the volumetric one-MP tax led to SSB purchase reductions close to 19%. Our inferred price elasticity of demand for SSB ranging -1.2 and -1.3 is similar to previous estimates of this elasticity equal to -1.16 based on income and expenditure data [40] or -0.9 based on countries with an SSB tax in place [39]. Our slightly larger findings of price elasticity of demand for SSB compared to previous studies might arise from our model accounting for both consumers and producers and by addressing the price endogeneity in the demand model. In their analyses of the literature on price elasticity determinants, Bijmolt et al. [53] found an increase in the absolute value of price elasticity when accounting for the price endogeneity.

Our findings suggest that one-MP equivalent sugar-density taxes will be better suited to tackle the high prevalence of overweight and obesity in Mexico compared to the existing volumetric one-MP tax as long as these sugar-density taxes encourage reformulation. Specifically, when assuming an exogenous reformulation linked to the sugar density taxes, we found that the one-MP equivalent sugar-density taxes outperformed the effect on sugar compared to the volumetric one-MP tax. This larger tax effect on sugar was particularly evident for the ZA tax design because we assumed reformulation across all SSB given the existing beverages in Mexico having sugar density above four grams per 100 ml. Conversely, we only assumed reformulation under the UK tax design across those few SSB that were right above the sugar-tax thresholds. In light of the prevailing sugar distribution in Mexico's SSB market, the ZA's threshold-linear sugar-density design is likely to lead to larger sugar reductions in Mexico by encouraging a more pronounced reformulation compared to the UK's multi-tiered sugar-density design.

Parallel to the sugar reduction under reformulation scenarios for the one-MP equivalent sugar density taxes, we found a tax revenue drop. This drop results from a lower tax burden at the product level due to the reformulation, and thus smaller price increases that mitigate the tax effect on SSB purchases in terms of volume. Therefore, our findings confirm that larger tax effects on sugar under the reformulation attributable to sugar-density taxes would entail a trade-off with tax revenue. Specifically, we estimated the largest tax revenue under the volumetric one-MP tax, which by its design, does not encourage reformulation and the lowest tax revenue under the extreme case of sugar-reduction reformulation for the ZA tax design across all producers. Consequently, the question about the most appropriate SSB tax design in

Mexico will depend on whether the objective of the tax in the short term is to maximize SSB purchase reductions in terms of either volume or sugar or to maximize tax revenue. In Mexico, Basto-Abreu et al. [54] simulated that SSB reformulation, i.e., sugar reduction, will lead to future reductions in average body weight and thus reductions in the prevalence of obesity. Therefore, our findings regarding lower tax revenue under sugar-density taxes with reformulation should be weighed against longer-term implications of sugar reduction on health and its potential healthcare savings.

Given that a volumetric tax design is currently in place in Mexico, it is unlikely to change due to the administrative burden to revise implemented systems. Our results show that in the absence of reformulation, the current volumetric one-MP tax performs well relative to one-MP equivalent sugar-density tax designs, especially with regards to SSB reductions by volume. However, the most recent health and nutrition survey in 2018 shows that the national prevalence of overweight and obesity remains extremely high, reaching eight out of ten adults, and increases in the prevalence of diabetes [55]. Therefore, sugar-density taxes on SSB in Mexico might represent an appropriate tax design to tackle the public health problems linked to SSB in case these taxes successfully encourage reformulation, as shown in the UK and ZA [19, 20]. Moreover, the SSB tax re-design toward sugar-density taxes might reinforce the potential reformulation and subsequent sugar intake reduction linked to the front-of-packaging labeling policy for food and beverages in effect in Mexico since October 2020 [56].

## Conclusions

In this study, we simulated the effect of the UK and ZA sugar-density taxes on SSB purchases, SSB prices, and tax revenue in Mexico. We compared the effect of these taxes to the volumetric one-MP SSB tax in effect since 2014. Moreover, we accounted for the potential reformulation attributable to sugar-density taxes by assuming some exogenous and extreme reformulation scenarios. In the absence of reformulation, we found that international equivalent and one-MP equivalent sugar-density taxes respectively tended to yield either lower or similar effects on SSB purchases in terms of volume and sugar compared to the volumetric one-MP tax given the Mexican beverage market structure. However, under our reformulation scenarios, the one-MP equivalent sugar-density taxes outperformed the sugar reduction compared to the volumetric one-MP tax. We found the largest sugar reductions under the ZA threshold-linear sugar-density tax. From a public health perspective, sugar-density taxes are more effective in tackling the overweight and obesity prevalence in Mexico; however, the larger sugar reduction under these taxes entails a trade-off with tax revenue. Future efforts should estimate the health and cost-benefit implications across the different tax designs given these and other trade-offs to assess both short-term and longer-term cost-benefits.

## Supporting information

**S1 Fig. Monthly SSB price average in urban Mexico from January 2012 to December 2015.** Note: Prices are calculated as quantity-weighted average prices. Source: Authors' own analyses and calculations based on data from Nielsen through its Mexico Consumer Panel Service (CPS) for the food and beverage categories for January 2012 –December 2015. The Nielsen Company, 2016. Nielsen is not responsible for and had no role in preparing the results reported herein.
(TIF)

**S2 Fig. Monthly average marginal costs by tax status from January 2012 to December 2015.** Note: Average marginal costs for the full categories of taxed and untaxed beverages

calculated as quantity-weighted average marginal costs. Light red (blue) lines represent the marginal costs for individual taxed (untaxed) products. Source: Authors' own analyses and calculations based on data from Nielsen through its Mexico Consumer Panel Service (CPS) for the food and beverage categories for January 2012 –December 2015. The Nielsen Company, 2016. Nielsen is not responsible for and had no role in preparing the results reported herein. (TIF)

**S1 Table. First stage estimates for the price endogeneity.** Note: Model includes the sugar content variable and the brand fixed effects as controls. SE: Standard error. + p < 0.10, * p < 0.05, ** p < 0.01, *** p < 0.001 Source: Authors' own analyses and calculations based on data from Nielsen through its Mexico Consumer Panel Service (CPS) for the food and beverage categories for January 2012 –December 2015. The Nielsen Company, 2016. Nielsen is not responsible for and had no role in preparing the results reported herein. (DOCX)

**S2 Table. Model fit for the volumetric one-Mexican peso SSB tax in 2014 and 2015.** Note: Prices are calculated as quantity-weighted average prices. MP: Mexican pesos. Source: Authors' own analyses and calculations based on data from Nielsen through its Mexico Consumer Panel Service (CPS) for the food and beverage categories for January 2012 –December 2015. The Nielsen Company, 2016. Nielsen is not responsible for and had no role in preparing the results reported herein. (DOCX)

**S3 Table. Sensitivity analyses for the demand model.** Note: Standard error in parentheses. Model includes brand fixed effects. Kid stands out for the presence of household members aged <13 years. + p< 0.10, * p < 0:05, ** p < 0:01, *** p < 0:001. Source: Authors' own analyses and calculations based on data from Nielsen through its Mexico Consumer Panel Service (CPS) for the food and beverage categories for January 2012 –December 2015. The Nielsen Company, 2016. Nielsen is not responsible for and had no role in preparing the results reported herein. (DOCX)

**S4 Table. Differential tax effects on SSB prices and purchases for the UK tax design.** Note: Prices are calculated as quantity-weighted average prices. MP: Mexican Pesos, PC-PD: Per capita Per Day. Source: Authors' own analyses and calculations based on data from Nielsen through its Mexico Consumer Panel Service (CPS) for the food and beverage categories for January 2012 –December 2015. The Nielsen Company, 2016. Nielsen is not responsible for and had no role in preparing the results reported herein. (DOCX)

**S5 Table. Tax effect in 2014 and 2015: Reformulation scenario of sweetness unchanged with null marginal cost adjustments.** Note: Prices are calculated as quantity-weighted average prices. MP: Mexican Pesos, PC-PD: Per capita Per Day. Source: Authors' own analyses and calculations based on data from Nielsen through its Mexico Consumer Panel Service (CPS) for the food and beverage categories for January 2012 –December 2015. The Nielsen Company, 2016. Nielsen is not responsible for and had no role in preparing the results reported herein. (DOCX)

**S1 File. Supplemental appendix.**
(DOCX)

## Acknowledgments

We wish to thank Dr. Donna Miles for exceptional assistance with the data management, Emily Yoon and Bridget Hollingsworth for administrative assistance.

## Author Contributions

**Conceptualization:** Juan Carlos Salgado Hernández, Shu Wen Ng.

**Data curation:** Juan Carlos Salgado Hernández, Shu Wen Ng.

**Formal analysis:** Juan Carlos Salgado Hernández.

**Funding acquisition:** Shu Wen Ng.

**Investigation:** Juan Carlos Salgado Hernández, Shu Wen Ng.

**Methodology:** Juan Carlos Salgado Hernández, Shu Wen Ng.

**Project administration:** Shu Wen Ng.

**Resources:** Shu Wen Ng.

**Software:** Juan Carlos Salgado Hernández.

**Supervision:** Shu Wen Ng.

**Validation:** Juan Carlos Salgado Hernández, Shu Wen Ng.

**Visualization:** Juan Carlos Salgado Hernández.

**Writing – original draft:** Juan Carlos Salgado Hernández, Shu Wen Ng.

**Writing – review & editing:** Juan Carlos Salgado Hernández, Shu Wen Ng.

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
