## [Decision Letter · Decision Letter 0]

7 Jan 2021

PONE-D-20-36090

Simulating international tax designs on sugar-sweetened beverages in Mexico

PLOS ONE

Dear Dr. Salgado Hernández,

Thank you for submitting your manuscript to PLOS ONE. After careful consideration, we feel that it has merit but does not fully meet PLOS ONE’s publication criteria as it currently stands. Therefore, we invite you to submit a revised version of the manuscript that addresses the points raised during the review process.

We look forward to receiving your revised manuscript.

Kind regards,

Guillermo Paraje, Ph.D

Academic Editor

PLOS ONE

Journal Requirements:

2. In the ethics statement in the manuscript and in the online submission form, please provide additional information about the data used in your retrospective study.

Specifically, please ensure that you have discussed whether all data were fully anonymized before you accessed them.

Reviewers' comments:

Reviewer's Responses to Questions

**Comments to the Author**

1. Is the manuscript technically sound, and do the data support the conclusions?

Reviewer #1: Yes

Reviewer #2: Partly

2. Has the statistical analysis been performed appropriately and rigorously? 

Reviewer #1: Yes

Reviewer #2: No

3. Have the authors made all data underlying the findings in their manuscript fully available?

Reviewer #1: Yes

Reviewer #2: No

4. Is the manuscript presented in an intelligible fashion and written in standard English?

Reviewer #1: Yes

Reviewer #2: Yes

5. Review Comments to the Author

Reviewer #1: This is an important article that examines alternative sugar-sweetened beverages tax scenarios in Mexico. The authors suggest that a sugar-density tax, which encourages product reformulation, would better tackle obesity in Mexico by producing a larger effect in demand than the volumetric tax. The results from this study are relevant throughout the Latin American region and elsewhere, as many countries are implementing strategies to reduce the consumption of these beverages. I do not have major comments regarding the methodology. However, I have suggestions about topics that should be added to the discussion section. In addition, I have a few minor comments throughout the manuscript.

1. The analysis is focused on Mexican urban household purchases, considering the 20-liter jars of plain drinking water as an “outside option”. The market share for this type of product has been found to be different among urban versus rural households. I think this should be acknowledged in the discussion section as this could have implications in the interpretation of the models.

2. Authors acknowledge that the current tax in Mexico is unlikely to change. Meanwhile Mexico has recently implemented a front-of-packaging labeling policy which is likely to encourage beverage reformulation. It think the authors could include a few sentences in the discussion about the possible synergy between these two strategies on demand and consumption of SSBs.

3. I think it would be worth to include a sentence of how the authors defined SSBs and non-SSBs for the purpose of this study. Were they defined according to a grams/100 mL threshold? Added sugar content? Any sugar content?

4. Page 6, Line 111: It would be worth to include a line saying that this panel captures the decline in sugar content on beverages after 2014 tax implementation

5. Page 9, line 175: I suggest that the authors review the manuscript for consistency on how the one-Mexican Peso volumetric tax is referred to throughout the manuscript and tables

6. Page 11, line 212: I suggest that authors only use abbreviations if the terms will be mentioned more than 3 times throughout the manuscript to improve readability

7. Page 15, Figure 2: I recommend that the authors re-label the figure to indicate that the Y-axis refers to Percentage of beverage

8. Page 17, Table 2: Please use a more clear label to refer to the “Kid” coefficient in the table. Alternatively, the authors could explain this label on a footnote

9. Page 18, line 326: Results suggest a negative marginal cost for plain water and authors suggest that it is possible that producers might be obtaining a subsidy or credit to produce plain water. I suggest that they elaborate on this and provide a reference

Reviewer #2: Referee report on

“Simulating international tax design on sugar-sweetened beverages in Mexico”,

By Juan Carlos Salgado & Shu Wen Ng.

This research article aims at simulating the impact of various tax schemes on the Mexican market for non-dairy and non-alcoholic beverages. The authors leverage household scanner data to calibrate a structural econometric model of supply and demand. The scanner data spans the period 2012-2015 that includes the implementation of one Mexican Peso (MP) tax on all beverages containing less than 1g of sugar per L. They first use the estimated model to simulate the counterfactual market equilibrium that would prevail if Mexico had implemented the recent UK and South-African tax schemes that make the tax rate function of the sugar content of beverages (with a progressive linear scheme for South-Africa and a discrete scheme distinguishing low- and high-sugar products for the UK). They find little difference with the estimated impacts of the one-MP tax: the reductions in quantities and sugar intakes are slightly lower (around -15% vs. -18% for the one-MP tax). They then explore the consequences of reformulation strategies that producers would be likely to implement if sugar content were taxed. Two strategies are simulated (1) a substitution of sugar ingredients with artificial sweeteners that would leave unchanged the taste of products (2) a mere reduction in sugar content. They find much larger reduction in sugar intakes under these scenarios (about -40% to -45%).

Overall, I find that the research question is interesting and important for the public health community. Soft-drink taxes have now been adopted in many jurisdictions, with different tax designs, market structures and consumer preferences. Hence, any comparative approach can help identify the best tax schemes in terms of public health, consumer welfare and distributional effects. This study has therefore the potential to contribute to the literature. Its main strength is that the authors implements up-to-date techniques to model the beverage market.

However, I have important concerns and suggestions regarding the internal validity of the empirical analysis, the scenarios that are simulated and the discussion of the results.

Empirical model

The authors adopt an Empirical Industrial Organisation (EIO) approach that has been used inter alia in studies of soft-drink taxes (Bonnet and Réquillart 2013) and nutritional labeling (Allais, Etilé et al. 2015). In simplified terms, to model a market, one must identify a demand curve and a supply curve. A key issue is that the supply curve is difficult to identify directly because data on firms’ production technologies, marginal production costs and margins are often unavailable. Berry, Levinsohn et al. (1995) proposed to overturn this problem through a two-step approach. Demand functions for the various products are estimated first, using available market data (prices and market shares). Then supply functions are calibrated by assuming that the market is in equilibrium and has a specific structure in terms of competition. Here, the authors assume a Bertrand-Nash oligopolistic competition between a limited set of producers (as in Allais et al., 2015), and they do not model the vertical relationships between retailers and producers (unlike, e.g., Bonnet & Réquillart, 2013). The main target parameters for the supply functions are the unit costs of production, which are identified by inverting the first-order conditions of profit maximisation by the producers. The authors implement this approach using a random-coefficient logit (RCL) model for modelling consumer demand (available in a Stata package, see Vincent, 2015).

The main threat to the internal validity of this approach is that the demand functions may be mis-specified and are difficult to identify. I now comment more precisely on these points:

1. The demand model assumes that the relevant market for Sugar-Sweetened Beverages (SSB) includes non-alcoholic non-dairy beverages and takes 20-liter jars of plain drinking water as the outside option:

a. The products are essentially differentiated in terms of brand (product fixed-effect in equation (1)) and sugar-content. However, the sugar-content appears to be product-specific and invariant over time, so that its marginal utility cannot be identified separately from the product fixed-effect. The only way to achieve identification is to have a brand fixed-effect (e.g. Coca-Cola) and introduce additional product attributes such as the sugar content (e.g. Light/Zero vs. regular) or the packaging, see Dubois, Griffith et al. (2020).

b. However, the sugar-content may have varied over time, following actual product reformulations by producers. The paragraph (P6 L109-112) discussing the nutritional information obtained at the product level should be more precise about this: what is the “average sugar content at the product level”? (P6, L111)

c. Do we have any information that may help to justify the exclusion of non-dairy products from the relevant market for SSB? Do urban households have access to tap water?

d. I understand that the price of the outside option is set at 0 (P7 L138). Were the product prices specified in terms of difference with the price of the outside option?

2. I understand that a market is one month: the identification does not attempt to exploit cross-sectional variations, e.g variations in market structure between Mexican cities. In addition, the model is estimated using the data aggregated at a monthly level, not the individual-level data.

a. Would it be possible to estimate directly mixed logit models of household choices (without aggregation)? Or it does not make sense because the survey is constructed to be exploited at an aggregate level only?

b. The estimated standard-deviation of random price coefficient is close to zero and not significant. This is not surprising because the identification of the random-coefficient distribution in RCL relies essentially on price variations that must be large or rather uncorrelated between products (see for instance assumption (12) in Fox, il Kim et al. (2012)). Here, the two main source of identifying variations are the one-MP tax and the time variations in sugar costs. Hence, the price variations that are relevant for identification are likely to be correlated across products. In addition, the within-product price variations are not large.

c. It would therefore be useful to complete this analysis by the estimation of standard logit model, to check the stability of the estimated price and sugar coefficients. This standard logit model can be estimated by regressing the log market shares (log(s_jt)-log(s_0t)) on the covariates, controlling for product fixed-effects. Note that this shows that identification of the taste for sugar requires that sugar-content display variations within products. Otherwise, it drops when one controls for the product fixed effects.

d. One way to have more variability is to leverage variations between cities. As local markets have different structures, there is a variability that is helpful for instrumenting the prices and for identification (see Etilé, Lecocq et al. (2020) for evidence on market heterogeneity in soft-drink tax incidence).

e. More powerful instruments may perhaps be obtained by interacting the sugar content of products at baseline (in 2012) with the time variations in the producer price index for sugar.

3. A key identifying assumption is that the one-MP tax did not affect consumer tastes. Hence, the tax increase can be used as a strong instrument for prices.

a. This is a debatable assumption that should be discussed in the presentation of the demand model (P7) and in the discussion of the identification strategy (P8). A soft-drink tax may impact consumer tastes by making the health risks of SSB consumption more salient and, over the long-term, by denormalising soft-drinks. These non-price effects are likely to be observed primarily in the upper classes (see the example of Tobacco)

b. An alternative strategy is to use the pre-tax data to calibrate the model, and then compare its predictions under the one-MP, the UK and the South-African scenarios. The authors should test the robustness of their main results to a switch to this alternative strategy. This would also be a means of conducting a proper test of model fit: does the model predict accurately what happened after the implementation of the one-MP tax?

4. Miscellaneous points:

a. The presentation of the inference method (P8 L147-152) lacks clarity. The important element in the approximation (3) is that the household demographics and unobservable characteristics are drawn simultaneously. From my own experience, the number of draws – here set to 500 – can have an impact on the estimated parameters. It is therefore necessary to provide evidence that increasing the number of draws to 1,000, 2,000, 5,000 does not improve the precision of the estimates.

b. I do not understand the role of the seasonal dummies. If they have the same effects for all utilities, then they cannot have an impact on the relative market shares. Or do these dummies capture the idea that the relative choice of the outside option vs. the inside products depends on season?

Scenarios

5. Many economists would apply a counterfactual scenario with fiscal neutrality, therefore asking what the tax rates under the UK (or ZA) tax design would be if one wanted to have the same tax revenues (assuming no consumer/producer price reactions) as in the one-MP scenario. Interestingly, this seems to be what the one-MP scenario without reformulation is achieving (this could be mentioned on P22, L381-382).

6. In the “sweetness unchanged” reformulation scenario, the authors assume that marginal costs increase by some non-negligible amount. I am not certain that the marginal cost would really be affected by such a reformulation. There would certainly be fixed costs (sourcing, R&D and perhaps marketing to launch the reformulated product). Then, an increase in marginal cost would depend on whether artificial sweeteners are more expensive than sugar-based sweeteners. Can the authors document this point?

7. The sweetness unchanged scenario assumes that artificial sweeteners and sugar have similar effects on consumer experience: this is a strong assumption (and the authors implicitly acknowledge that P12, L247-248).

a. In principle, it could be possible to estimate a specific taste parameter for artificial sweeteners. As there are only 30 products, it is not difficult to find how their recipes evolve over time. Accounting for this is likely to affect the results as we know that consumers are attached to certain ‘iconic’ recipes (e.g Coca-Cola regular).

b. The analysis of trends in soft-drink markets shows that producers of retailer brands (private labels) and minor national brands are much more likely to reformulate their products than market leaders. The latter tend rather to launch new products with a healthy image (e.g tea-based drinks) or light equivalent to their regular products, or they extend their product portfolio by buying healthier brands. This should be discussed in the Discussion section.

c. There are also some studies that have examined the impact of product dynamics on nutrition - see Griffith, O'Connell et al. (2017), Spiteri and Soler (2018). They show that products entry can sometimes limit the impact of the reformulation of existing products, because consumers tend to stick to their tastes for sugar, fat and salt. This outlines the importance of identifying separately the taste for sugar and the taste for artificial sweeteners.

8. It would be interesting to decompose more formally the effect of the tax + reformulation scenarios into three components: (1) a tax-induced price effect holding composition constant; (2) a change in sugar content at post-tax price level holding post-tax market shares constant; (3) the change in post-tax market shares (at post-tax price levels) that may be induced by the change in demand following reformulation. This third component thus identifies the likely reaction of consumers to reformulation. It is null under the current “sweetness unchanged” scenario but it would not be null if consumers had a specific (dis)taste for artificial sweeteners.

Other points

9. A potential drawback of discrete choice models is that the market size is fixed or varies exogenously (Q_t in equations A.2.2 and A.2.1.). This is probably a reasonable assumption in the Mexican context and given that water is the outside option. But this should be discussed somewhere.

10. The instrumental variables only capture shocks on the prices of SSB products. Hence, the estimated marginal utility of income (the price coefficient) will essentially reflect the variations in SSB market shares in response to price shocks. The estimates may underweight the price response of diet products and water. This should be mentioned in the discussion.

Allais, O., F. Etilé and S. Lecocq (2015). "Mandatory labels, taxes and market forces: An empirical evaluation of fat policies." Journal of health economics 43: 27-44.

Berry, S., J. Levinsohn and A. Pakes (1995). "Automobile prices in market equilibrium." Econometrica: Journal of the Econometric Society: 841-890.

Bonnet, C. and V. Réquillart (2013). "Tax incidence with strategic firms in the soft drink market." Journal of Public Economics 106(0): 77-88.

Dubois, P., R. Griffith and M. O'Connell (2020). "How well targeted are soda taxes?" American Economic Review 110(11): 3661-3704.

Etilé, F., S. Lecocq and C. Boizot-Szantai (2020). "Market heterogeneity and the distributional incidence of soft-drink taxes: evidence from France." European Review of Agricultural Economics.

Fox, J. T., K. il Kim, S. P. Ryan and P. Bajari (2012). "The random coefficients logit model is identified." Journal of Econometrics 166(2): 204-212.

Griffith, R., M. O'Connell and K. Smith (2017). "The importance of product reformulation versus consumer choice in improving diet quality." Economica 84(333): 34-53.

Spiteri, M. and L.-G. Soler (2018). "Food reformulation and nutritional quality of food consumption: an analysis based on households panel data in France." European journal of clinical nutrition 72(2): 228-235.

6. PLOS authors have the option to publish the peer review history of their article (what does this mean?). If published, this will include your full peer review and any attached files.

Reviewer #1: **Yes: **Violeta Chacón

Reviewer #2: **Yes: **Fabrice Etilé

---

## [Author Response · Author response to Decision Letter 0]

23 Mar 2021

Please, see the file for responses to reviewers

---

## [Decision Letter · Decision Letter 1]

21 Apr 2021

PONE-D-20-36090R1

Simulating international tax designs on sugar-sweetened beverages in Mexico

PLOS ONE

Dear Dr. Salgado Hernández,

Thank you for submitting your manuscript to PLOS ONE. After careful consideration, we feel that it has merit but does not fully meet PLOS ONE’s publication criteria as it currently stands. Therefore, we invite you to submit a revised version of the manuscript that addresses the points raised during the review process.

We look forward to receiving your revised manuscript.

Kind regards,

Guillermo Paraje, Ph.D

Academic Editor

PLOS ONE

Journal Requirements:

Reviewers' comments:

Reviewer's Responses to Questions

**Comments to the Author**

1. If the authors have adequately addressed your comments raised in a previous round of review and you feel that this manuscript is now acceptable for publication, you may indicate that here to bypass the “Comments to the Author” section, enter your conflict of interest statement in the “Confidential to Editor” section, and submit your "Accept" recommendation.

Reviewer #2: (No Response)

2. Is the manuscript technically sound, and do the data support the conclusions?

Reviewer #2: Yes

3. Has the statistical analysis been performed appropriately and rigorously? 

Reviewer #2: Yes

4. Have the authors made all data underlying the findings in their manuscript fully available?

Reviewer #2: No

5. Is the manuscript presented in an intelligible fashion and written in standard English?

Reviewer #2: Yes

6. Review Comments to the Author

Reviewer #2: The authors did a great job in dealing with most of my comments. I have only four comments left:

- I am not fully satisfied with the answer to my comment 2b about the estimated standard-deviation of random price coefficient, which is not significant and close to zero: the authors consider that this implies that “households observed characteristics drive price heterogeneity in utility”. This is rather far-fetched, as the choice of working on aggregate market shares and the choice of IVs explain this result. Please reformulate appropriately and add a footnote to mention the identification issue (with the reference to Fox, Kim et al).

- S1 Table, computation of the first-stage F-stat: given the legend of the Table, it is not clear to me that you included the sugar content variable, and the product fixed-effects as controls. Please revise the table and/or the statistics appropriately.

- About the inclusion of seasonal dummies that do not vary across inside options: this is definitely not standard in the empirical IO and in the quantitative marketing literature. Hence, you should better justify their inclusion in your main text: why would we expect specific seasonal shifts in the taste for inside options vs. the outside option?

- Please provide in the Appendix the table of the marginal costs by brand j that were estimated from the structural supply-side model. This is important for ascertaining the reliability of the model.

7. PLOS authors have the option to publish the peer review history of their article (what does this mean?). If published, this will include your full peer review and any attached files.

Reviewer #2: **Yes: **Dr. Fabrice Etilé (Paris School of Economics, INRAE Research Professor)

---

## [Author Response · Author response to Decision Letter 1]

3 Jun 2021

Please see the file on responses to reviewers

---

## [Editor Report · Decision Letter 2]

14 Jun 2021

Simulating international tax designs on sugar-sweetened beverages in Mexico

PONE-D-20-36090R2

Dear Dr. Salgado Hernández,

We’re pleased to inform you that your manuscript has been judged scientifically suitable for publication and will be formally accepted for publication once it meets all outstanding technical requirements.

Kind regards,

Guillermo Paraje, Ph.D

Academic Editor

PLOS ONE
---

## [Editor Report · Acceptance letter]

29 Jul 2021

PONE-D-20-36090R2 

Simulating international tax designs on sugar-sweetened beverages in Mexico 

Dear Dr. Salgado Hernández:

I'm pleased to inform you that your manuscript has been deemed suitable for publication in PLOS ONE. Congratulations! Your manuscript is now with our production department. 

Kind regards, 

on behalf of

Dr. Guillermo Paraje 

Academic Editor

PLOS ONE